# CapS-Adapter: Caption-based MultiModal Adapter in Zero-Shot Classification

## ABSTRACT

Recent advances in vision-language foundational models, such as CLIP, have demonstrated significant strides in zero-shot classification. However, the extensive parameterization of models like CLIP necessitates a resource-intensive fine-tuning process. In response, TIP-Adapter and SuS-X have introduced training-free methods aimed at bolstering the efficacy of downstream tasks. While these approaches incorporate support sets to maintain data distribution consistency between knowledge cache and test sets, they often fall short in terms of generalization on the test set, particularly when faced with test data exhibiting substantial distributional variations. In this work, we present CapS-Adapter, an innovative method that employs a caption-based support set, effectively harnessing both image and caption features to exceed existing state-of-the-art techniques in training-free scenarios. CapS-Adapter adeptly constructs support sets that closely mirror target distributions, utilizing instance-level distribution features extracted from multimodal large models. By leveraging CLIP's single and cross-modal strengths, CapS-Adapter enhances predictive accuracy through the use of multimodal support sets. Our method achieves outstanding zero-shot classification results across 19 benchmark datasets, improving accuracy by 2.19% over the previous leading method. Our contributions are substantiated through extensive validation on multiple benchmark datasets, demonstrating superior performance and robust generalization capabilities.

## CCS CONCEPTS

• **Computing methodologies → Computer vision tasks**.

## KEYWORDS

Vision-Language Models, CLIP, Training-free, Multimodal Support Set

## 1 INTRODUCTION

Recent advancements in vision-language foundation models (VLMs) [17, 21, 29] have marked significant progress across various computer vision tasks. These models exhibit strong zero-shot capabilities, having been pretrained on large-scale image-text pairing datasets, one prominent example of them is CLIP. When applying VLMs to downstream tasks, if the data distribution of the downstream dataset differs significantly from the image distribution used

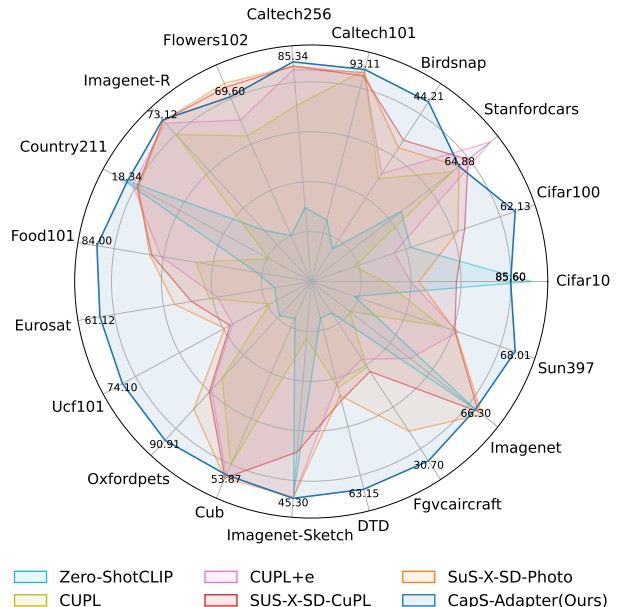

**Figure 1: Radar chart. The line in the color ▭ represents our method _CapS-Adapter_. _CapS-Adapter_ demonstrates superior performance on 19 datasets.**

during VLMs' pre-training, its zero-shot performance substantially decreases [10].

Therefore, some studies aiming at adapting VLMs for diverse downstream tasks have been introduced in previous works. These methods primarily fall into four categories: manual prompts adjustment, prompt learning methods [41, 42], feature tuning methods [24, 39], or training-free methods [33, 40]. Among these, manual prompts require human knowledge and effort to create, and their effectiveness is often limited [42]. On the other hand, while prompt learning and feature tuning methods adapts by fine-tuning on a subset of the target task's data, its highly parameterized nature makes these methods prone to instability and an inherent tendency to overfit [7, 12]. To address this, training-free methods have been introduced and shown to be effective. They introduced a knowledge cache for downstream tasks, formed by a collection of images. This collection is referred to as the "support set" by SuS-X [33]. However, due to the high similarity and a lack of instance-level information in SuS-X, the constructed support set deviates from the target distribution. This deviation leads to a decrease in the method's performance as the number of images in the support set increases. Therefore, exploring more effective and general methods for constructing support sets is considered crucial. Moreover, previous training-free methods often solely utilize the image features of the support set, which leads to a focus on the intra-modal correlations between test set and the support set. However, vision-language models like CLIP possess both cross-modal and intra-modal capabilities. From

this perspective, incorporating text features into the support set, formatted as image inputs, is meaningful. Previously, SuS-X proposed using a text classifier to bridge image-image correlations, transforming them into image-text-image correlations. However, introducing such intermediary bridges is less intuitive than directly incorporating relevant text features into the support set and considering image-text correlations outright. Therefore, exploring the use of multimodal support sets that include text features is important.

To address these issues, we propose **CapS-Adapter** in this paper, which adjusts vision-language models for downstream classification tasks in a *training-free* manner. Specifically, the **CapS-Adapter** approach is divided into two parts. *(1)* The first component is the *CapS* (Caption-based Support Set), a *multimodal support set* that is closely aligned with the target distribution, along with an efficient method for its construction. This system utilizes a multimodal large language model to generate captions for a small subset of images sampled from the target distribution training set. These captions contain instance-level semantic information. Subsequently, these image captions are blended with category texts to create caption-based prompts. These prompts are then input into a large-scale text-to-image generation model (e.g., Stable Diffusion), resulting in a diverse set of support images that match the target distribution. The CLIP similarity between these images and the target distribution's test set improved by an average of 1.5% over baseline methods. The features of these images and the caption-based prompts together form this caption-based *multimodal support set*, providing a knowledge cache for zero-shot classification. *(2)* Building upon our constructed *CapS*, we propose the *M-Adapter* (Multimodal-Adapter), a method for tailoring visual language models to downstream tasks using *CapS*. It leverages features from both the images in *CapS* and the caption-based prompts. By calculating the association matrix $A_M$, it adeptly balances text-image cross-modal similarity and image-image intra-modal similarity for downstream prediction. The *M-Adapter* effectively utilizes the multimodal features within the support set, and even with identical images in support set, it outperforms sota (state-of-the-art) method SuS-X by 1.22% in performance. As shown in **Figure** 1, **CapS-Adapter** boosts classification performance across 19 benchmark datasets with average accuracy increases of 5.28%, 2.28% and 2.19% respectively.

The contributions of this paper are as follows:

- We propose a novel support set, *CapS*, and its construction method, which innovatively incorporates textual information into the support set. By effectively utilizing instance-level information from image captions, it generates more generalized downstream representations. It addresses previous issue where performance declined as the number of images in the support set increased.
- For the *CapS* architecture, we introduced *M-Adapter*, an inference approch that optimally leverages cached multimodal features during the classification process. This method is training-free.
- Our approach, **CapS-Adapter**, which combines *CapS* with *M-Adapter*, achieves state-of-the-art results, outperforming previous method by 2.19% in a training-free scenario on 19 datasets.

## 2 RELATED WORK

### 2.1 Vision-Language Models (VLMs)

Visual language models demonstrate strong performance across a range of visual tasks and possess powerful generalization capabilities, such as CLIP[29], a model trained on a vast dataset of text-image pairs through contrastive learning. This approach has since inspired a plethora of visual language models that employ similar training methodologies. The pre-trained CLIP model acquires the ability to represent images and text in a shared feature space through contrastive learning. These image-text representations derived from CLIP can be utilized for downstream tasks like semantic segmentation and object detection. Notably, CLIP demonstrates the capability to handle zero-shot classification in these tasks by employing *class prompts* in the form of "A photo of <CLASS>."

### 2.2 VLMs' Adaptation

Inspired by the zero-shot ability of CLIP, subsequent work aims to improve its performance. The ability of CLIP to handle zero-shot classification in downstream tasks is influenced by the data distribution of those tasks. Many researchers have proposed methods for downstream task adaptation in response to this issue, enhancing CLIP's capabilities on specific downstream task distributions through prompt learning or training-free methods.

*2.2.1 Prompt Learning.* The Context Optimization (CoOp) [42] method, by converting context words in *class prompts* into a set of learnable vectors, introduces the trend of prompt learning from the NLP domain into the vision domain, achieving significant performance improvements with a small number of labeled images, surpassing intensively-tuned manual *class prompts*. However, CoOp exhibits an overfitting issue with classes observed during training, and its generalization to unseen categories within the same dataset is limited. To address this issue, the Conditional Context Optimization (CoCoOp) [41] method was proposed, extending CoOp by learning a lightweight neural network to generate an input-conditional token (vector) for each image. Compared to the static prompts used in CoOp, CoCoOp's dynamic prompts adapt to each instance, reducing sensitivity to class shifts. Experimental results show that CoCoOp outperforms CoOp in generalizing to unseen classes, even demonstrating promising transferability across different datasets, while also providing stronger domain generalization performance. But the issue of overfitting continues to be present in enhanced prompt-learning methods such as CoOoOp.

*2.2.2 Training-free Methods.* Some methods that require no learning leverage few-shot approaches, using a small number of samples from the training set as a knowledge cache available for reference during inference. These methods incorporate the image features of the samples into the inference process of computing logits, thus enhancing the zero-shot capabilities of CLIP.

SuS-X [33] employs a "name transfer only" method, which leverages the category names and the concepts of categories understood by large language models. This method generates a series of prompts by GPT-3 [4] and constructs a support set through Stable Diffusion [30] generation and LAION-5B [31] retrieval, achieving state-of-the-art performance. However, this method is constrained

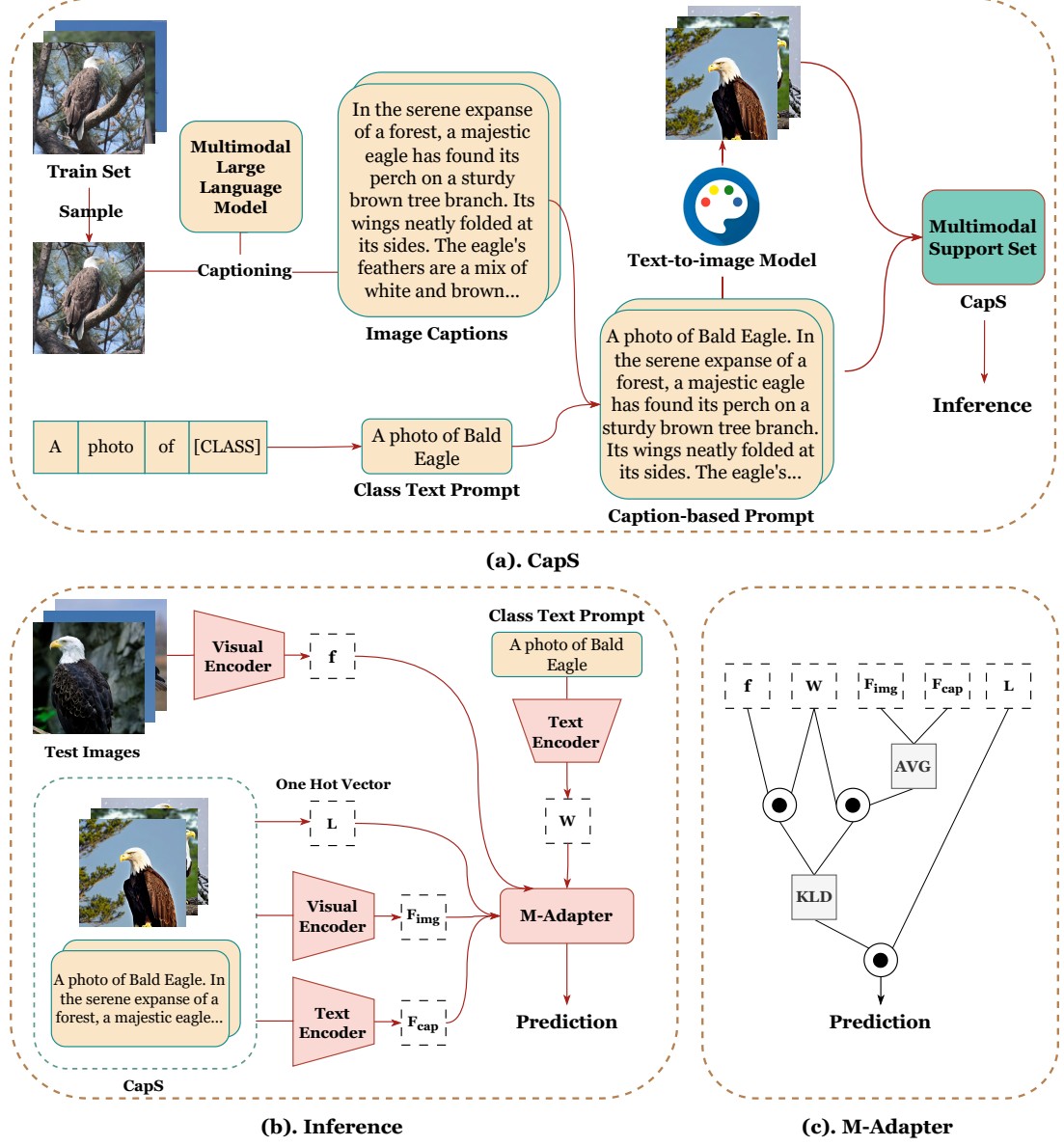

**Figure 2:** *Caps-Adapter* workflow. (a)*CapS*. It utilizes the image captions and category text as prompts. These prompts are used with a text-to-image model to create diverse images. These images and captions together form the *CapS*. (b) Utilizing the zero-shot *M-Adapter* for inference, which leverages the image and caption features from *CapS* to generate predictions. (c) Details of *M-Adapter*. It integrates the caption, category text, and image features to generate the similarity between the test images and categories.

by the knowledge of large language models. The prompts generated by large language models often focus on common-sense text, lacking consideration for uncommon, niche domains. Moreover, these prompts lack instance-level semantic information, resulting in the support sets generated by this method often exhibiting significant discrepancies in data distribution compared to the target dataset images across many datasets. This leads to a high degree of similarity and redundancy in the information contained within the images of the support sets.

## 2.3 Multimodal Large Language Models

The integration of MLP adapters to project encoded image features into the input feature space of Large Language Models (LLMs) and similar methods have led to the emergence of numerous Multimodal Large Language Models (MLLMs) with powerful image comprehension and linguistic capabilities [5, 20, 22, 23, 37]. The latest advancements in Multimodal Large Language Models (MLLMs) demonstrate their powerful capabilities in generating detailed and contextually relevant captions for images. A notable contribution in this field is ShareCaptioner [5], an open-source model fine-tuned

with the assistance of GPT4-Vision [1], capable of producing accurate and richly detailed captions by fine-tuning on an image-caption pair dataset with rich details.

## 3 METHOD

The overall process of our method is shown in **Figure** 2. To overcome the gap between the support set built in previous training-free methods and target distribution, we designed a *multimodal support set* named *CapS* and method to construct it, as shown in **Figure** 2 (a). We constructs *CapS* based on image captions. On top of *CapS*, we designed an inference approach for prediction. It uses features of both image and text modal in *CapS*, named *M-Adapter*. It addresses the issue of not fully leveraging VLMs' cross-modal capabilities when solely using the image features of the support set.

### 3.1 *CapS*: Caption-based Multimodal Support Set

The latest training-free adaptation method employs a set of images to provide CLIP with visual knowledge for downstream tasks. This image set is named *support set*. We leverages image captions to develop the multimodal support set *CapS*. Our method considers the instance-level features in captions, thus the images in the generated support set are more closely aligned with the target distribution. We innovatively incorporated caption-based prompts, which contains textual features, into the support set. *CapS* is structured around two key components: caption-based prompts and generated images.

*3.1.1 Generate Caption-based Prompts.* We utilize a multimodal large language model to obtain image captions. We concatenae image captions with class text prompts to obtain *caption-based prompts*. Specifically:

Given a downstream task dataset containing $N$ categories, our objective is to create a *multimodal support set* as a cache tailored for the downstream task, incorporating instance-level knowledge of $N$ categories. For each category in the training set, we extract $K$ images, denoted as $I_K$, and input these images into a multimodal large language model (MLLM) named ShareCaptioner[5], to obtain captions for these images, for the $i_{th}$ image $I_i$, its caption $C_i$ is

$$C_i = \Omega(I_i). \tag{1}$$

For all $NK$ samples, their captions are denoted as $C_{NK}$. $\Omega$ means multimodal large language model. Leveraging the image interpretation and summarization capabilities of multimodal large language models, $C_{NK}$ encompasses information on the data distribution of the downstream task in textual form.

For each class in $N$ categories, the *class text prompt* we use is a very simple sentence "A photo of <*classname*>." For a special datasets Country211 , the prompt is another simple category prompt, "In <*classname*>." The *class text prompt* for $N$ classes are denoted as $P_N$, which contains class information about the downstream task. By concatenating prompts in $P_N$ and $C_{NK}$, we obtain the Caption-Based Prompt (*CBP*), denoted for the $j_{th}$ image in $i_{th}$ class as

$$CBP_{ij} = \text{concat}(P_i, C_{ij}). \tag{2}$$

The concatenated $CBP_{NK}$ includes both the instance Level information obtained from image captions and the category information

from *class text prompts*. It will be used to generate the image part of the support set.

*3.1.2 Image Generation.* We utilized the text-to-image model, *Stable Diffusion*, to accomplish image generation. For k-th class, randomly samples of its *caption-based prompt*, $CBP_K$, is used as input of *Stable Diffusion* to generate a collection of $M$ images, $I_M$. Since these $M$ prompts are randomly selected from $CBP_K$, duplication of prompts occurs when $M > K$. To avoid repetition in $I_M$ when $M > K$, we use different random seeds in *Stable Diffusion* generation for same *caption-based prompt*.

*3.1.3 Multimodal Support Set.* Subsequently, we constructed a *multimodal support set*, *CapS*. For $N$ classes, *CapS* involves integrating the collection of *caption-based prompts*, $CBP_{NM}$, with the generated images, $I_{NM}$. When we need to access the cached knowledge in *CapS*, it is necessary to encode the images and text within *CapS*:

For each image in $I_M$, we employ a pre-trained CLIP visual encoder to extract its image features. Similarly, for each caption-based prompt in $CBP_M$, we utilize the CLIP text encoder to extract its text features. Both the image and text features have a dimensionality of $C$. For all $NM$ images, the encoded visual features are denoted as $F_{img} \in \mathbb{R}^{NM \times C}$,

$$F_{img} = \text{CLIPEncoder}_{\text{visual}}(I_M). \tag{3}$$

Likewise, for all $NM$ caption-based prompts, the encoded text features are represented as $F_{cap} \in \mathbb{R}^{NM \times C}$

$$F_{cap} = \text{CLIPEncoder}_{\text{text}}(CBP_M). \tag{4}$$

### 3.2 *M-Adapter*: Inference Approach

Based on *CapS* constructed previously, we propose a training-free inference approach, *M-Adapter*, to enhance the prediction capabilities of zero-shot CLIP in downstream tasks. In this section, we will introduce the classification inference method for zero-shot CLIP, which serves as the foundation for a series of improvement efforts, and our *M-Adapter*.

*3.2.1 Zero-shot CLIP.* For a classification task comprising $N$ categories, the prediction process of zero-shot CLIP initially involves transforming category labels into text prompts, typically crafted manually. The most fundamental text prompt used for zero-shot CLIP predictions is the *class text prompt* "A photo of <*classname*>." Subsequently, these text prompts and the images to be classified are encoded into features in the feature space of CLIP using a pre-trained encoder. The *M-Adapter* is shown in **Figure** 2(c).

The feature of one single image to be tested is denoted as $f_{\text{test}} \in \mathbb{R}^{1 \times C}$, where $C$ represents the dimension of the feature. Similarly, for a batch of $t$ test images, their features are represented as $f_{\text{test}}^t \in \mathbb{R}^{t \times C}$. The text feature vectors are aggregated into a CLIP classifier $W \in \mathbb{R}^{N \times C}$, with $N$ being the number of classes.

Compute the dot product of $f_{\text{test}}$ and $W$ to obtain the similarity logits between $f_{\text{test}}$ and the prompt feature of each class,

$$\text{logits} = f_{\text{test}} \cdot W^T. \tag{5}$$

The logits are then used to yield the zero-shot CLIP prediction results, by taking the label of maximum value in the logit vector for each test image.

**Table 1: Main results. We compare the classification accuracy of *CapS-Adapter* with other training-free methods and zero-shot CLIP across 19 benchmark datasets. The data presented are the average results from experiments conducted on five CLIP backbone networks (ResNet-50, ResNet-101, ViT-B/32, ViT-B/16, and ViT-L/14), with detailed results for each backbone network provided in the appendix. On each dataset, the best and second-best results are indicated in bold and underlined, respectively. \*Avarage is calculated across 19 datasets.**

| | Birdsnap | CIFAR-10 | CIFAR-100 | CUB | Caltech101 | Caltech256 | Country211 | DTD | EuroSAT | FGVCAircraft | Flowers102 | Food101 | ImageNet | ImageNet-R | ImageNet-Sketch | OxfordPets | SUN397 | StanfordCars | UCF101 | Average* |
|---|---|---|---|---|---|---|---|---|---|---|---|---|---|---|---|---|---|---|---|---|
| ZS-CLIP | 35.65 | **85.8** | 59.40 | 46.77 | 90.95 | 83.97 | **18.38** | 45.39 | 39.18 | 21.31 | 66.62 | 81.12 | 66.29 | 71.75 | 45.26 | 85.2 | 63.43 | 64.62 | 62.47 | 59.66 |
| CuPL | 39.73 | 84.39 | 57.93 | 53.40 | 93.10 | 84.93 | 17.36 | 52.63 | 47.38 | 24.75 | 68.76 | 82.21 | 62.89 | 72.94 | 43.88 | 88.07 | 65.92 | 64.85 | 62.96 | 61.48 |
| CuPL+e | 40.04 | 84.64 | 58.97 | 53.87 | 93.07 | 85.27 | 18.30 | 52.23 | 46.36 | 24.25 | 69.10 | 82.85 | 64.53 | 73.08 | 44.85 | 88.60 | 66.38 | **65.02** | 65.77 | 61.96 |
| SUS-X-SD-Photo | 41.54 | 84.72 | 60.53 | 53.43 | 93.02 | 85.30 | 18.27 | 53.94 | 51.76 | 24.82 | **69.89** | 83.05 | **66.47** | 73.11 | 45.29 | 89.48 | 66.31 | 64.88 | 66.37 | 62.75 |
| SUS-X-SD-CuPL | 41.98 | 85.08 | 60.81 | **54.01** | 93.02 | 85.30 | 18.27 | 53.94 | 49.63 | 25.03 | 69.81 | 83.01 | 66.38 | 73.11 | 45.30 | 88.71 | 66.29 | 64.92 | 65.96 | 62.66 |
| CapS-Adapter (Ours) | **44.21** | 85.6 | **62.13** | 53.87 | **93.11** | **85.34** | 18.34 | **63.15** | **61.12** | **30.70** | 69.60 | **84.00** | 66.30 | **73.12** | 45.30 | **90.91** | **68.01** | **65.02** | **74.10** | **64.94** |

*3.2.2 M-Adapter.* The *M-Adapter* is an improved inference method based on the TIP-X[33] from SuS-X. The workflow of *M-Adapter* is shown in **Figure** 2(c). TIP-X adapts CLIP for zero-shot tasks by incorporating image-label caching, matrix-vector multiplication, and KL divergence. Specifically, it enhances the zero-shot framework by introducing two additional terms: $\alpha AL$ and $\gamma\varphi(-ML)$, where:

$$\text{logits} = f_{\text{test}}^t W^T + \alpha AL + \gamma\varphi(-ML). \tag{6}$$

$f_{\text{test}}^t \in \mathbb{R}^{t \times C}$ represents the feature vector. $L$ denotes the one-hot vector matrix converted from labels. **A** and **M** are the association and intimacy matrices introduced by TIP-Adapter and SuS-X, respectively.

Matrix **A** calculates the association between the test image (considered as a query) and the pre-computed feature vectors of image-label pairs:

$$A = \exp(-\beta(1 - f_{\text{test}}^t \cdot F_{img})). \tag{7}$$

$\beta$ is an adjustable hyperparameter that modulates the "sharpness," making **A** more sensitive to variations in $f_{test}$ and $F_{img}$. $\alpha$ in $\alpha AL$ is the residual ratio when mixing this term with zero-shot predictions.

**M** utilizes the zero-shot CLIP text classifier as a cross-modal bridge to represent the affinity within the same modality between $f_{test}$ and $F_{img}$, calculated through the KL divergence between two signatures $s_i$ and $S_j$:

$$M_{i,j} = KL(s_i \| S_j), \tag{8}$$

for $i \in [1, t]$ across $t$ test images, and $j \in [1, CM]$ across $M$ images in the support set, with $C$ denoting the feature dimension.

Before constructing matrix **M**, it is necessary to compute two signatures $S \in \mathbb{R}^{CM \times C}$ and $s \in \mathbb{R}^{t \times C}$, representing the similarities between the text classifier weights $W$ and $f_{test}^t$, and $W$ and $F_{img}$, respectively:

$$S = \text{softmax}(F_{img} W^T), \tag{9}$$

$$s = \text{softmax}(f_{test}^t W^T). \tag{10}$$

After calculating **M**, an automatic scaling function $\varphi$ adjusts **M** to align its value range with that of **A**. $\gamma$ in $\gamma\varphi(-ML)$ is the residual ratio for mixing this term with others.

Addressing the issue of large variance in CLIP's intra-modal similarity scores, TIP-X utilizes the zero-shot CLIP text classifier

as an intermediary bridge. Building on TIP-X, *M-Adapter* modifies the inclusion of the support set's feature cache by incorporating both image feature and caption feature (text feature) caches. This is achieved by calculating the weighted mix of similarities between $f_{test}^t$ and the cached features, leading to a new association matrix **A**$_M$ (M for *multimodal*):

$$A_M = \exp(-\beta(1 - \delta f_{test}^t \cdot F_{cap} - (1 - \delta) f_{test}^t \cdot F_{img})). \tag{11}$$

$\delta$ is a newly introduced hyperparameter adjusting the balance between text-image cross-modal similarity and image-image modal similarity in **A**$_M$, with larger $\delta$ values indicating a greater emphasis on the similarity between the support set's stored text features and the test images.

We still use $\alpha$ and $\gamma$ as the hyperparameters to mix the terms in the logits, *M-Adapter* is represented as

$$\text{logits} = f_{test}^t \cdot W^T + \alpha A_M L + \gamma\varphi(-ML), \tag{12}$$

where **A**$_M$ is defined by **Equation** 11.

## 4 EXPERIMENTS

### 4.1 Experimental Settings

We evaluated the comparison results of ***Caps-Adapter*** against baselines across 19 widely-used image classification datasets, targeting the training-free adaptation scenario of the visual language model CLIP: Birdsnap [2], Caltech101 [11], Caltech256 [13], Cifar10 [19], Cifar100 [19], Country211 [29], CUB [35], DTD [6], Eurosat [15], FGVCAircraft [25], Flowers102 [26], Food101 [3], ImageNet [8], ImageNet-R [16], ImageNet-Sketch [36],OxfordPets [27], Stanford-Cars [18],SUN397 [38], and UCF101 [32].

We compared the performance with three zero-training methods: zero-shot CLIP [29], CuPL [28], and SuS-X [33]. For zero-shot CLIP, we utilized seven prompt templates [29, 40] to generate text classifiers. We ran CuPL and SuS-X using their official code. In addition to this, for CuPL, we executed its mixed variant CuPL+e, following the implementation in SuS-X, which combines it with the seven prompt templates used in the seven zero-shot CLIP scenarios. Classified by the approach to obtaining support sets, SuS-X is implemented in two ways: the retrieval method SuS-X-LC and the generative

(a) Apple Pie                  (b) Arctic Tern

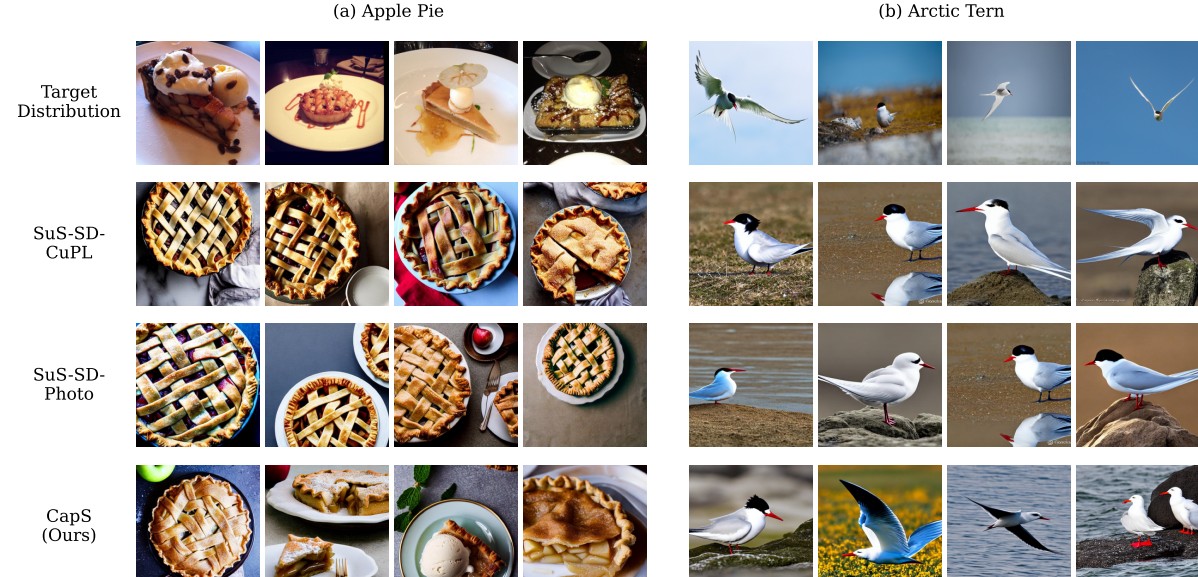

**Figure 3: Data sampled from target distribution and support set images of SuS-SD-CuPL, SuS-SD-Photo, *CapS*. Image samples *CapS* are more diverse and closer to the target distribution: showcasing a variety of apple pie shapes and both dynamic and static images of arctic terns.**

method SuS-X-SD. The capabilities of these two methods are very similar. Each implementation method can also be divided into CuPL mode (GPT3-generated) and Photo mode(manually constructed), according to the prompt mode used when querying or generating images. Since our method employs Stable Diffusion to generate images for constructing the support set, we considered two results for SuS-X in our report: SuS-X-SD-Photo and SuS-X-SD-CuPL. For the prompt mode of the text classifier in the SuS-X reasoning process, we have predominantly utilized the combined mode, which exhibits superior performance on the majority of datasets. However, for ImageNet and ImageNet-Sketch, we have employed the ensemble mode, which performs better on these two datasets specifically. In order to make a strict comparison with SuS-X, the prompt mode of the text classifier in the CapS reasoning process is kept identical to SuS-X.

Previous no-learning adaptation methods primarily used ResNet-50 [14] as the image encoder for CLIP. We believe that only considering a single CLIP backbone network is insufficient to fully reflect the performance of adaptation methods. Therefore, we conducted experiments using five CLIP backbone networks as encoders: ResNet-50, ResNet-101, ViT-B/32, ViT-B/16, and ViT-L/14 [9]. We reported the average results across these five backbone networks for each dataset in the main text and provided the complete results for each backbone network in the supplementary materials.

## 4.2 Main Result

Our experiments and analyses across all 19 datasets, as shown in **Table** 1, demonstrate that *CapS-Adapter* significantly outperforms other training-free methods. Across all 19 datasets, the *CapS-Adapter* approach achieves a 5.28% improvement on average over

the zero-shot CLIP, as well as an average improvement of 2.28% and 2.19% over SuS-X-SD-CuPL and SuS-X-SD-Photo, respectively.

Specifically, among the listed six training-free methods, *CapS-Adapter* achieved the highest accuracy in 14 out of 19 datasets and the second-highest accuracy in 3 datasets. Furthermore, we discovered that *CapS-Adapter* excels in several fine-grained classification datasets. Compared to zero-shot CLIP, improvements on the EuroSAT, DTD, UCF101, FGVCAircraft, and Birdsnap datasets were 21.94%, 17.76%, 11.63%, 9.39%, and 8.56%, respectively, improvements over SuS-X-SD-Photo were 9.36%, 9.21%, 7.73%, 5.88%, and 2.67%, and improvements over SuS-X-SD-CuPL were 11.49%, 9.21%, 8.14%, 5.67%, and 2.23%, respectively.

As shown in **Table** 1 that the *Caps-Adapter* significantly enhances performance on datasets involving fine-grained classification and uncommon category classification, such as Birdsnap (birds), EuroSAT (satellite images), DTD (textures), UCF101 (actions), FGVCAircraft, and Food101, compared to the baseline method SuS-X. We attribute these significant improvements primarily to the datasets' heightened sensitivity to the quality of image features within the support set. The superior quality of image features in *Caps* is mainly because the image categories in these datasets are not widely represented in the pre-training of text-to-image generation models like Stable Diffusion, which lack sufficient prior knowledge about these categories. Consequently, the generation of support set images relies heavily on the input prompts. *Caps* utilizes caption-based prompts, which offer a well-distributed, rich, and varied instance-level information compared to the simpler GPT-3 generated or manual prompts used by SuS-X, thus better guiding the support set image generation process. The widespread improvement across 19 datasets is attributed to the *M-Adapter*'s efficient utilization of caption text features in *Caps*, in contrast to SuS-X,

**Table 2: Ablation Study. We compared the classification accuracy of SuS-SD-Photo+TIP-X (SuS-X-SD-Photo), SuS-SD-CuPL+TIP-X (SuS-X-SD-CuPL), *CapS*+TIP-X, and *CapS*+*M-Adapter* on 19 datasets, reflecting the effects of *CapS* and *M-Adapter*. The best and second-best results are indicated in bold and underlined respectively. *Avarage is calculated across 19 datasets.**

| | Birdsnap | CIFAR-10 | CIFAR-100 | CUB | Caltech101 | Caltech256 | Country211 | DTD | EuroSAT | FGVCAircraft | Flowers102 | Food101 | ImageNet | ImageNet-R | ImageNet-Sketch | OxfordPets | SUN397 | StanfordCars | UCF101 | Average* |
|---|---|---|---|---|---|---|---|---|---|---|---|---|---|---|---|---|---|---|---|---|
| SuS-SD-Photo+TIP-X | 41.54 | 84.72 | 60.53 | 53.43 | 93.02 | 85.30 | 18.27 | 53.94 | 51.76 | 24.82 | **69.89** | 83.05 | **66.47** | 73.11 | 45.29 | 89.48 | 66.31 | 64.88 | 66.37 | 62.75 |
| SuS-SD-CuPL+TIP-X | 41.98 | 85.08 | 60.81 | **54.01** | 93.02 | 85.30 | 18.27 | 53.94 | 49.63 | 25.03 | 69.81 | 83.01 | 66.38 | 73.11 | **45.30** | 88.71 | 66.29 | 64.92 | 65.96 | 62.66 |
| CapS(Ours)+TIP-X | 42.12 | 85.09 | 61.42 | 53.97 | 93.00 | 85.31 | 18.34 | 61.85 | 55.71 | 27.43 | 69.56 | 83.09 | 64.59 | 73.10 | 44.91 | 89.90 | 67.23 | 64.83 | 69.14 | 63.72 |
| CapS+M-Adapter (Ours) | **44.21** | **85.60** | **62.13** | 53.87 | **93.11** | **85.34** | 18.34 | **63.15** | **61.12** | **30.70** | 69.60 | **84.00** | 66.30 | **73.12** | **45.30** | **90.91** | **68.01** | **65.02** | **74.10** | **64.94** |

which only utilizes image features of the support set during inference. Further analysis of the effects of *Caps* and *M-Adapter* is presented in our ablation study.

## 5 ABLATION STUDY

***CapS-Adapter*** consists of two modules: the support set module *CapS* and the inference module *M-Adapter*. To analyze the effects of these two components, we conducted ablation studies. These studies involved experiments on 19 datasets using image part of *CapS* and the inference module *TIP-X* from the baseline method SuS-X. The results of the experiment are shown in **Table 2**. The results are compared with those of *CapS*+*M-Adapter* (***CapS-Adapter***), SuS-X-SD-Photo (SuS-SD-CuPL+TIP-X), and SuS-X-SD-CuPL (SuS-SD-Photo+TIP-X). Given the high degree of integration between *M-Adapter* and *CapS* (with *M-Adapter* relying on the multimodal knowledge cache within *CapS*), and the absence of a textual feature knowledge cache in SuS-SD, we did not conduct experiments on SuS-SD with *M-Adapter*.

### 5.1 Effects of Caption-based Multimodal Support Set (*CapS*)

*5.1.1 Data Distribution Analysis.* *CapS* aims to address the issue of image distribution deviation in the support set constructed by previous methods from the target data distribution. This section will focus on this aspect, comparing the data distribution of the support sets constructed by the *CapS* method and the SuS-X-SD method.

**Figure** 3 presents randomly sampled image examples from two data categories corresponding to the target test set distribution, the support set images generated by SuS-SD, and the support set images from *CapS*, specifically for the Apple Pie from the Food101 dataset and the Arctic Tern from BirdSnap. The pictures generated by the two SuS-SD generation modes exhibit characteristics that are somewhat repetitive and deviate from the target distribution. For instance, their samples for the Apple Pie category in **Figure** 3(a) primarily display the round shape of apple pies, and in **Figure** 3(b), their samples for the Arctic Tern category only show static images of the arctic tern. In contrast, in *CapS*, thanks to the instance-level features introduced by caption-based prompts, the image distribution is closer to the target distribution, with the samples in **Figure**

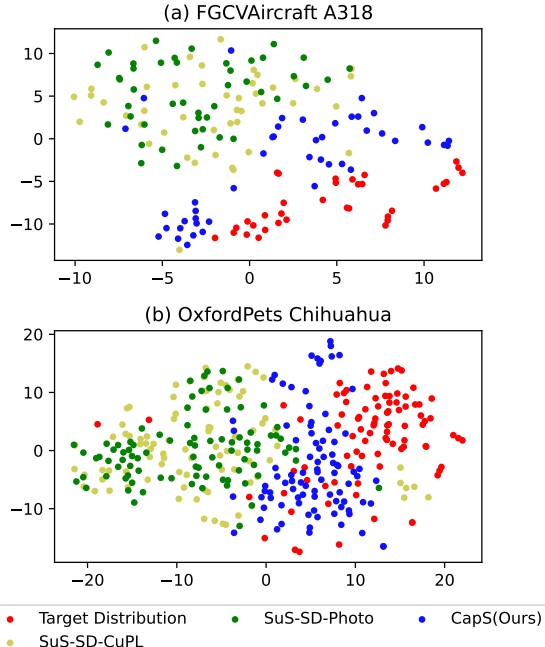

(a) FGCVAircraft A318

(b) OxfordPets Chihuahua

Target Distribution    SuS-SD-Photo    CapS(Ours)    SuS-SD-CuPL

**Figure 4: Data distribution comparison. Visualized image features of samples from the the Target Distribution, support sets generated by SuS-SD-CuPL, SuS-SD-Photo, and image part in *CapS*. Features from *CapS* are notably closer to the target distribution and more diverse.**

3 showcasing a variety of apple pie shapes and both dynamic and static images of arctic terns.

We randomly sampled 50 images each from the target test set distribution corresponding to the A318 class in the FGCVAircraft dataset, the support set image distribution generated by SuS-SD, and the support set image distribution from the image part of *CapS*. Similarly, we sampled 100 images each from these data distributions for the Chihuahua class in the OxfordPets dataset. These images were encoded using a pretrained CLIP visual encoder, then dimensionality reduction was performed using t-SNE[34] for visualization. In **Figure** 4, the visualized features show that the image features of SuS-SD-Photo and SuS-SD-CuPL are more concentrated

**Table 3: Comparison of CLIP similarity(%) between images in support set and target test set. The CLIP similarity performance of *CapS* is better. Results on other datasets are provided in the appendix. \*Avarage is calculated across 19 datasets.**

| Method | Birdsnap | Food101 | OxfordPets | UCF101 | Average* |
|---|---|---|---|---|---|
| SuS-SD-CuPL | 67.77 | 64.93 | 84.97 | 54.83 | 69.93 |
| SuS-SD-Photo | 68.20 | 66.10 | 88.08 | 57.43 | 71.14 |
| CapS(Ours) | **79.94** | **79.12** | **94.66** | **70.86** | **72.64** |

and distant from the features of the target distribution, reflecting the characteristic that the images in their support sets are relatively homogeneous and deviate from the target distribution. On the other hand, the image features of *CapS* are closer to the target distribution features while being more dispersed, reflecting their notably closer proximity to the target distribution and greater diversity.

To evaluate whether the image distribution of the support sets closely resembles the target data distribution, we adopted the method of calculating the average CLIP similarity between the images in the support set and the test set of the target dataset. This metric was calculated for the support sets constructed for all 19 datasets, with results for Birdsnap, Food101, OxfordPets, UCF101, and the average results across the 19 datasets illustrated in **Table** 3 (detailed results for each of 19 datasets are available in the supplementary materials). The average CLIP similarity between the images in *CapS* and the dataset test sets was found to be 1.5% and 2.71% higher than that of SuS-SD-CuPL and SuS-SD-Photo, respectively.

*5.1.2 Performance Analysis.* From rows 1-3 of **Table** 2, it is evident that using image part of *CapS* enhanced the performance of the baseline method across most datasets, resulting in an average accuracy increase of 0.97% and 1.06%. This indicates that CapS's approach to generating support set images indeed produces collections of images with a more favorable data distribution, providing a more effective knowledge cache for zero-shot classification.

Researchers if SuS-X posit that providing more support set samples is always beneficial when the true data distribution closely resembles that of the support set samples [33]. However, when there is a significant discrepancy between the two, increasing the number of image samples in the support set can be counterproductive. It can be inferred that the effectiveness of the support set is reflected by changes in method performance as the number of support set image samples varies. To this end, we selected scenarios with support set image counts of 5, 10, 25, 50, 75, and 100, and visualized the changes in classification accuracy for four methods—***CapS-Adapter***, *CapS* + TIP-X, SuS-X-SD-Photo, and SuS-X-SD-CuPL—across these counts in the datasets FGVCAircraft and SUN397, as shown in **Figure** 5.

From the images in rows 1-3 of **Figure** 5, it can be observed that when using SuS-SD as the support set, TIP-X's performance on FGVCAircraft and SUN397 tends to decline as the number of support set images increases. In contrast, replacing SuS-SD with the image part of *CapS* reverses this trend, resulting in improved performance with an increase in the number of images. This demonstrates that the image part of *CapS* is more closely aligned with the true data distribution and effectively enhances method performance.

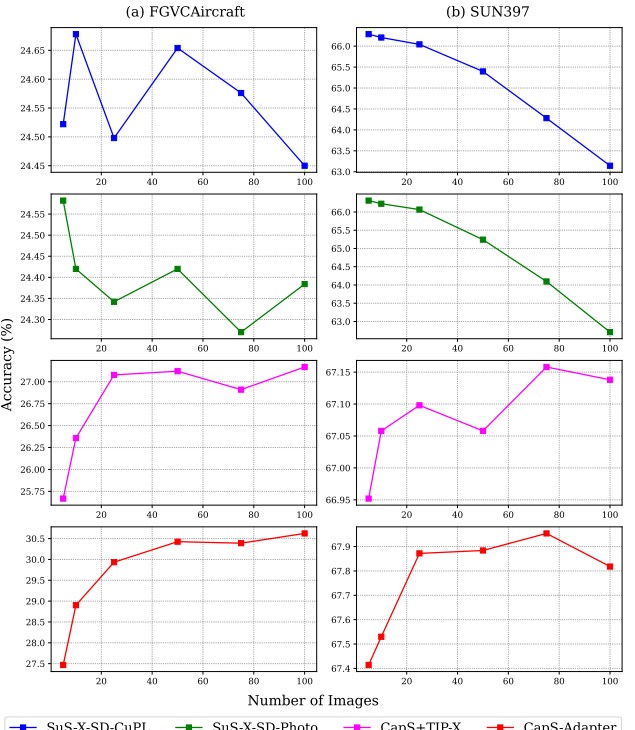

**Figure 5: Accuracy changes as the number of images in the support set increases.**

## 5.2 Effects of Multimodal Adapter (*M-Adapter*)

*M-Adapter* plays a critical role in the ***CapS-Adapter*** by simultaneously considering both text and image features from *CapS* during the inference process. As illustrated by rows 3 and 4 in **Table** 2, when using *CapS*, incorporating M-Adapter at inference outperformed the baseline method TIP-X[33] in 18 out of 19 datasets, with an average improvement of 1.22%. This demonstrates that M-Adapter's multimodal approach to inference more effectively utilizes the knowledge cache stored in the support set compared to TIP-X, which only leverages image features of the support set. The substantial improvement in row 4 over row 3 in **Figure** 5 also corroborates this finding.

## 6 CONCLUSION

This paper introduces *CapS-Adapter*, a pioneering training-free approach in the domain of vision-language models' adaptation, which successfully addresses the limitations of existing training-free methods. By leveraging a unique caption-based support set, *CapS-Adapter* effectively utilizes both image and text features, closely approaching the target distributions, and demonstrates superior performance in zero-shot classification tasks over previous state-of-the-art methods. This achievement highlights the potential of integrating multimodal support sets to achieve robust generalization capabilities, emphasizing the effectiveness of instance-level distribution features and multimodal data handling in enhancing predictive outcomes.

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
