# OpenReview forum: "CapS-Adapter: Caption-based MultiModal Adapter in Zero-Shot Classification"
_acmmm.org/ACMMM/2024/Conference — MM2024 Poster_

### Official Review · Reviewer_Xk9y · 2024-05-20

**Rating:** 4
**Confidence:** 3

**Summary:**

The manuscript introduces CapS-Adapter ,a caption-based multi modal adapter in zero-shot classification designed to overcome a key challenge: the limited generalization capabilities of training-free methods when faced with substantial distributional variations in test data.

CapS-Adapter proposes two innovative modules, Caps (Caption-based Support Set) and M-Adapter (Multimodal-Adapter), to address potential challenges. CapS aims to build a multimodal support set that is tightly integrated with the target distribution. M-Adapter leverages features from both the images in CapS and the caption-based prompts, adeptly balances text-image cross-modal similarity and image-image intra-modal similarity for downstream prediction.

Experiments conducted on 19 dataset exhibited the method’s performance.

**Strengths:**

-The motivation of the paper is clear, and the proposed method is interesting as it takes into account certain problem in the zero-shot classification.

-The author explains the advantages and disadvantages of other works. Derive the two proposed modules with precise language and clear logical flow.

-The experiments are sufficient and comprehensive, demonstrating the effectiveness of the proposed method.

**Limitations:**

-Major
Regarding several hyperparameters α, β and γ mentioned in the paper, there seems to be no  specific values and corresponding ablation experiments.

-Minor
Reference: Some references are not formatted correctly (most conferences and journals names  have only abbreviations).

**Suitability:**

2

---

### Official Review · Reviewer_uKXH · 2024-05-21

**Rating:** 3
**Confidence:** 4

**Summary:**

Recent advances in vision-language foundational models like CLIP have shown promising results in zero-shot classification, but they are often hampered by their need for resource-heavy fine-tuning due to extensive parameterization. In response, novel training-free methods such as TIP-Adapter and SuS-X have been developed, yet they struggle with generalization in the face of significant test data distributional variations. The proposed CapS-Adapter method addresses these challenges by utilizing a caption-based support set that leverages both image and caption features, thus better mirroring target distributions and improving predictive accuracy. This method employs support sets derived from instance-level distribution features in multimodal large models and capitalizes on CLIP's modal strengths to enhance zero-shot classification performance. CapS-Adapter has demonstrated its effectiveness by surpassing previous methods on 19 benchmark datasets, achieving a notable 2.19% increase in accuracy and showcasing robust generalization across diverse datasets.

**Strengths:**

1. The problem of generalization is important in the current context of VLMs, and using supplementary caption knowledge is interesting.
2. The paper is overall well-written and mostly easy to follow.
3. Results are exhaustive in general.

**Limitations:**

. Line 153: Improvement in average accuracy - Please clarify what the improvement is related to.
2. The caption in Figure 2 does not properly define the variables. The variable "M" is confusing; consider using different variables for clarity. Additionally, the use of the AVG operator between F_img and F_cap is not mentioned in the model description, and the motivation for using AVG is unclear.
3. The authors should reference more recent studies on prompt learning, such as Prograd and MAPLe. Additionally, the results should be compared with other baselines such as CoOp, CoCoOp, CLIP-Adapter, and TipAdapter.
4. Clarify what is meant by "considering five backbones as the encoders." Were these used in separate experiments or ensembled into one architecture? Also, mention if CLIP’s image and text encoders are frozen in the figure and text. Please clarify the same in the experimental setup. How does the length of the generated captions affect the performance? How do we ensure the similarity between the generated images and support sets?

5. The caption in Table 3 states the average is across 19 datasets, but the values are associated with 4 datasets. Please explain this discrepancy.
6. Did you consider negative prompts for generating new images using Stable Diffusion? Provide more details on the type of stable diffusion used.
7. The T-SNE plots for the classes should be discussed in the paper.
8. The paper lacks ablation studies related to the different kinds of prompts and components used in Figure 2.
9. The authors have not considered several other benchmark methods, raising concerns about the average results reported in Table 1. While these results surpass the referenced methods, the models have not significantly outperformed the benchmark methods across all 19 datasets. This limitation could have been addressed by the proposed CapS-Adapter, which uses samples from the target distribution to generate related captions for zero-shot classification on the target scenes. Additionally, the authors should compare their proposed caption-based image generation with ODG-CLIP(CVPR 24)  under closed-set scenarios.

**Suitability:**

3

---

### Official Review · Reviewer_VueC · 2024-05-27

**Rating:** 4
**Confidence:** 3

**Summary:**

The paper propose CapS-Adapter, a training-free method for enhancing zero-shot classification performance using caption-based support sets. By leveraging both image and caption features, CapS-Adapter constructs support sets that closely resemble target distributions, effectively harnessing the strengths of vision-language foundational models like CLIP. Extensive validation on 19 benchmark datasets demonstrates CapS-Adapter's superior performance and robust generalization capabilities, achieving a 2.19% improvement in accuracy over the previous state-of-the-art method.

**Strengths:**

The idea of using a muti-modal support set and muti-modal adapter looks technically sound.

The performance improvement is clear.

**Limitations:**

1. The technique contribution is quite limited. Using caption rather than fixed prompt template in muti-modal learning has been widely explored in papers such as [1,2].

2.  The idea of the M-adapter is quite similar to [3], please illustrate and discuss it in the paper.

3. I am not very surprised about the performance improvement of the proposed method, as the overall idea looks like a combination of the advantages of [1][3][4], making the novelty of the work limited.

4. Does the additional muti-modal feature-based adapter learning cause additional inference costs than single-modal methods? Please add a comparison of inference cost.

5. A discussion on the difference between these works mentioned above is necessary in the paper.

[1 ]Generating customized prompts for zero-shot image classification

[2] Generalizing CLIP to Unseen Domain via Text-Guided Diverse Novel Feature Synthesis

[3] Multi-modal Prompt Learning

[4] Training-Free Name-Only Transfer of Vision-Language Models

**Suitability:**

3

---

### Official Review · Reviewer_rea3 · 2024-05-29

**Rating:** 3
**Confidence:** 3

**Summary:**

The authors present CapS-Adapter, an innovative method that employs a caption-based support set, effectively harnessing both image and caption features to exceed existing state-of-the art techniques in training-free scenarios.

**Strengths:**

1.The authors present CapS-Adapter, an innovative method that employs a caption-based support set, effectively harnessing both image and caption features to exceed existing state-of-the art techniques in training-free scenarios. (Training-Free setting)
2.CapS-Adapter adeptly constructs support sets that closely mirror target distributions, utilizing instance-level distribution features extracted from multimodal large models.
3. The method achieves outstanding zero-shot classification results across 19 benchmark datasets, improving accuracy by 2.19% over the previous leading method.

**Limitations:**

1. The overall framework could be treated as a combination of exploiting previous Large Models. CapS integrates the VLMs for captioning and then obtains the cap-prompts. The innovation in this part of the work is somewhat limited, and the author only enriched the expression of prompt through a large model.
2.The capability of M-adapter is mainly inherited from CLIP and inspired from  SuS-X. Novelty is limited.
3.The engineering nature of the article is strong, but its research is limited. As a zero shot oriented work, the author lacks analysis on improving domain gap discrimination. Although the author has conducted many validation experiments, they have indeed achieved good experimental results.
4. More Zero-shot methods based on improved CLIP should be compared.
5. Multi-modal may not suitable for this paper. Visual Language Model (VLM) is more suitable.

**Suitability:**

3

---

### Meta-Review · Area_Chair_1jNM · 2024-07-01

**Recommendation:** Accept (Poster)
**Confidence:** 4

**Metareview:**

The paper propose CapS-Adapter, a training-free method for enhancing zero-shot classification performance using caption-based support sets. By leveraging both image and caption features, CapS-Adapter constructs support sets that closely resemble target distributions, effectively harnessing the strengths of vision-language foundational models like CLIP. Validation on 19 benchmark datasets demonstrates its performance and generalization capabilities.

After the rebuttal, the paper receives 3 borderline accept and 1 borderline reject. The similarities to the related papers shall be better explained in the final version.